# Green Manufacturing Strategy Considering Retailers' Fairness Concerns

**Huanyong Zhang [1], Zhen Zhang [1,*], Xujin Pu [1] and Yuhang Li [2]**

[1]  Business School, Jiangnan University, Wuxi 214122, China
[2]  Business School, Nankai University, Tianjin 300071, China
*   Correspondence: littlejohnie@163.com; Tel.: +86-178-5130-8379

**Abstract:** This paper addresses the problem of green manufacturing decision making for a green dual-channel supply chain (SC). In the investigated SC, the manufacturer will decide whether to adopt green manufacturing under the influence of the retailer's fairness concern-based dual-channel. Thus, we discuss two decision scenarios: the no green manufacturing strategy with retailer fairness (*NM* model), and green manufacturing with retailer fairness (*GM* model). Our study has several findings: Firstly, adopting a green manufacturing strategy is not always beneficial to supply-chain members when a retailer has fairness. In particular, when fairness is at a relatively high level, the manufacturer will not adopt green manufacturing. Secondly, under green manufacturing, the product's green degree and subsidies have a positive impact on the price and demand and the members' profit and utility. Besides, the subsidies and retailer fairness have a counter effect on the optimal decision. Thirdly, comparing the two scenarios (*NM & GM*), we found that the channel price of the *GM* model is lower than the *NM* model. Finally, from the perspective of the supply chain system, the system tends toward the manufacturer adopting green manufacturing and maintaining retailer fairness concerns at a lower level.

**Keywords:** fairness concern; dual-channel; green manufacturing; pricing decisions

## 1. Introduction

With the development of the global economy, environmental issues have become the focus of the attention of the state, enterprises, and society in recent years. As a result, the importance of carrying out the *Green Economy*, characterized by being low-emission, low-energy, and low-pollution, has become a global consensus [1]. Green Supply Chain Management (GSCM) reconsiders all aspects of the supply system compared to traditional supply chains, involving SC design, production-related decisions, raw material procurement, etc. [2]. Enterprises, under the influence of various factors such as regulative and market pressure and the improvement of their competitive strength, need to transform their products and services into green products to meet environmental requirements and consumer needs. Numerous enterprises are also actively adopting green measures in response to the challenges posed by severe environmental problems and the requirements of green sustainable development advocated by various countries. For example, Adidas, Pepsi Cola, and Wal-Mart have adopted their supply strategies based on environmental and green concerns [3]. Besides, the European Environment Agency regularly publishes white papers to enhance consumers' environmental awareness. In 2008, SAIC General Motors launched a *Green Future* strategy, focusing on improvement of the environmental performance of suppliers and lean production. By 2015, SAIC General Motors had invested 420 million yuan in energy-saving and environmental protection improvement projects, from which enterprises have gained direct economic benefits of 380 million yuan. Global manufacturers are committed to adopting innovative and environmentally friendly technologies in the product design process to make

the supply chain green [4]. Besides, to manage their environmental footprint and that of their suppliers, Li & Fung have implemented a sustainable strategy. The significance of green supply chains and the increasing attention and investment paid to their development throughout the world have made it essential for both academicians and industry managers to consider these issues in their research and decision-making processes.

At the same time, the rapid development of technology and e-commerce has changed consumption patterns, with more and more consumers using online shopping. According to the 43rd internet network development statistics report in China, the number of people who use the Internet in China has reached 829 million, with over 610 million people shopping online, and the netizen usage rate reached 73.6% by the end of December 2018 [5]. Therefore, more and more enterprises want to expand their market share by building online direct sales channels (herein simplified to direct channel). However, the introduction of direct channels has exacerbated the conflicts of traditional retail channels. This kind of conflict will lead to changes in the original profit distribution of channel members. Fairness concerns refer to a firm's concern about the inequality between supply chain parties [6]. The uneven distribution of profits in the supply chain system often results in the fairness concern behavior of the members of the supply chain. Thus, when facing distribution problems, they are not only concerned about their personal income but also focus on whether they have been treated fairly in the allocated districts [7]. For example, in 2007, Longsha Group, China's largest sock manufacturer, terminated its cooperation with Wal-Mart over the unfair distribution of benefits. The *Wuchang rice phenomenon* is a problem caused by unfairness among members of the supply chain (the maximum selling price per kilogram of rice in Wuchang rice is 199 yuan, while rice farmers receive less than 2 yuan, and the processing cost is only 0.2 yuan).

In this paper, we develop a dual-channel supply chain consisting of one manufacturer and one retailer. The retailer has a fairness concern, and he pays attention to his profitability while comparing the manufacturer's earnings to reflect whether he is treated fairly in the supply chain system. The manufacturer compares the traditional dual-channel supply chains to make decisions about green manufacturing under the influence of the retailer's fairness concerns, channel competition, and government subsidies. The problems that the research project needs to solve can be summarized as follows: first, in contrast to previous research, when a retailer has fairness concerns, it is questioned whether the manufacturer adopting green manufacturing is beneficial to supply chain members and systems under the dual-channel mode. Second, under this strategy, what are the manufacturer's and retailer's optimal decisions? Third, how the retailer's fairness concerns impacts the the supply chain members' decisions when developing green manufacturing is investigated. Finally, we explore the effect of the product's green degree and government subsidies on optimal decision making. Thereby, the relationship between the manufacturer and retailer is depicted using a Stackelberg game, and a green dual-channel supply chain model is formulated to assist the manufacturer and retailer in making reasonable and optimal decisions.

The structure of the paper is organized as follows. Section 2 summarizes the related literature. In Section 3, based on the green dual-channel supply chain model, we describe the problem and present further assumptions and notations. Section 4 studies the optimal decision of members in the model. Section 5 verifies the applicability and effectiveness of the model by numerical analysis. Conclusions are presented in Section 6.

## 2. Literature Review

The three streams of research of green supply chains, dual-supply chain, and fairness concerns are closely related to this study. In this section, we provide a brief review of the relevant literature and highlight the differences between this study and the existing literature.

In recent years, some companies have actively extended their social responsibilities and actively tried to build green supply chains such as: increasing investment in green manufacturing, producing green products, and improving manufacturing processes. National manufacturing strategies such

as *German industrial 4.0* and *Made in China 2025* emphasize the importance of the development of green manufacturing. Li et al. [2] also proposed that green manufacturing plays a crucial role in GSCs. Many scholars have defined green manufacturing through their research. According to Deif [8], green manufacturing (GM) reflects a new manufacturing paradigm that incorporates various green strategies (objectives and principles), drivers (motivators and critical success factors) and techniques (technology and innovations) to become more eco-efficient. GM includes making or creating products/systems that consume less materials, less energy, substituting input materials (non-toxic for toxic, renewable for non-renewable) reducing unwanted outputs, wastes, emissions and converting outputs to in-puts (recycling). Pujari [9] pointed out that green product innovation could advance the green level, thus decreasing the environmental cost (i.e., improving environmental performance). Many studies have begun to focus on green manufacturing and green products. Wei et al. [10] analyzed the interaction between greening and remanufacturing strategies in a manufacturer—retailer supply chain by studying the manufacturer's collecting options of used products and decisions regarding the effort toward improving the green degree of its products under a dynamic scenario. Zhu et al. [11] investigated green product development issues in various competitive supply chain structures, including a coordinated supply chain, vertical competing supply chain, and horizontal competing supply chains. Ma et al. [12] explored the optimal pricing strategy for substitutable products for six different power structure models and discovered that green manufacturing will benefit the manufacturers involved in green investment. Hong et al. [13] studied the effects of consumers' reference behavior and environmental awareness on green-product design strategies for a two-echelon supply chain. Dey et al. [4] discussed the joint impact of a retailer's strategic decisions and consumers' continuous expectation on the investment and wholesale pricing decisions of the manufacturer in improving the green level of their product. Du et al. [14] studied the fairness of sustainable green technology innovation development by suppliers and manufacturers and found that whether the sustainable green technology innovation investments and profits for all the members are fair is a critical factor to motivate the supply chain members. In addition, some scholars have studied green supply chains from other dimensions, such as coordination, government regulation, and market competition. Hong et al. [15] designed three contracts, introduced them into the green product supply chain and investigated their environmental performance. Chen et al. [16] investigated the decisions of supply-chain members in green SC management under a reward–penalty mechanism from the government. Zhou et al. [17] investigated whether implementing a carbon tax policy would change the pricing decisions and social welfare of supply-chain members from the perspective of maximizing the social welfare of policy implementers. Hafezalkotob [18] considered the six regulation policies of deregulation, direct tariffs, direct limitations, government certificates, government permits, and cooperative energy saving and the competition of green supply chains. Some scholars have likewise studied the performance evaluation and coordination of green supply chains. Ma et al. [19] consider a dual-distribution green supply chain (GSC) to see how the alignment may be achieved through the competition and cooperation of the GSC participants. Li et al. [20] provided preliminary evidence regarding the impact of corporate quality management on green innovation and the moderating role of environmental regulations by analyzing the data of the top 100 listed companies from 2008 to 2014 in China. Al-Sheyadi et al. [21] examined the collective impact of internal and external GSCM practices on environmental impacts and environmental cost savings. From the above literature, only [19] is based on a dual-channel scenario, while [14] investigated the impact of fairness, and [16–18] are related to government subsidies or taxes. This paper contributes to the study of the impact of retailers' fairness concerns on green manufacturing strategies for manufacturers under a dual-channel system.

In addition, many manufacturers have sold their products by opening up direct online channels. As an example of the household appliance industry, Haier sells a series of green products, such as energy conservation refrigerators and air conditioners, through their own direct channel. Thus, in the research of dual-channel SCs, many scholars have begun to study dual-channel systems, and the literature can be divided into two streams: operation (i.e., the pricing decision, competition, and

coordination) under dual-channel and dual-channel structure strategies. Our model belongs to the former because we consider the impact of a retailer's fairness concerns on supply chain member decisions and the choice of green manufacturing strategy. In the first stream, Zhang et al. [22] considered an e-commerce retailer that offers online shopping and a dual-channel supply network. By introducing a dual-channel supply coordination issue, the e-commerce retailer can deliver products from its fulfillment center or vendor's warehouse based on the customer's geographic location. Wang et al. [23] addressed the problem that the reduction of carbon emissions is driven by cap-and-trade regulation and consumers' low-carbon preference in a dual-channel supply chain and analyzed the optimal pricing strategies and profits for supply chain members. Zhang et al. [24] investigated the choice of recycling strategy based on dual channels. They found that, for manufacturers, the decision regarding return policy depends only on whether the channel can save the returned product, and the competition between the channels will also have an impact; the opposite has occurred for brick-and-mortar retailers. Li et al. [25] considered a dual-channel supply chain where a manufacturer with a direct channel acts as the leader and a retailer exhibits fairness concerns. They found that channel efficiency grows with increasing customer loyalty to the retail channel and falls with increases in the retailer's fairness concerns. In the second stream, Yang et al. [26] investigated the dual-channel structure strategy of a green manufacturer by clearly characterizing the environmental responsibility behaviors of both the manufacturer and consumers and further examined its environmental performance under fuzzy uncertainties. Dumrongsiri et al. [27] constructed a two-channel supply chain model for manufacturers to initiate direct online marketing channels. The scholars found that manufacturers may be better off with two channels compared to a single channel when retailers had high marginal sales costs and low variability in wholesale prices, consumer values, and demand. From the above, in [22–27], the dual-channel system is discussed; only [25] studies fairness; in [22], government subsidy or regulation is investigated; and [23] investigated the green manufacturing of products.

In practice, fairness concerns are often the result of uneven distribution of members due to unequal market power, so the fairness of decision-making members has an impact on the distribution of channel profits and cooperation between them, such as the *Wuchang rice phenomenon* being a problem caused by unfairness among members of the supply chain (the maximum selling price per kilogram of rice in Wuchang rice is 199 yuan, while rice farmers receive less than 2 yuan, and the processing cost is only 0.2 yuan). Choi et al. [28] found the supply chain members tend to choose similar margin levels and profits which are more fairly divided than non-cooperative, game-theoretic, supply-chain models predicted by analyzing experimental results, which reflected individual supply chain members' behavior; this shows evidence of fairness concerns for supply chain members under different power structures. Cui et al. [6] was among the first to introduce the concept of fairness concerns into supply chain management research using an analytical model. They hold that fairness concerns refer to a firm's concern about the inequality between supply chain parties. For example, a manufacturer's different strategies for different retailers may cause retailers to perceive unfairness. Consequently, the manufacturer and retailer all present certain reactions (e.g., change pricing strategy, add an online direct channel or refuse to cooperate) to solve the unfairness or the fairness concern. In our work similar to this, we investigated the choice of a green manufacturing strategy under dual-channel when the retailer has fairness concerns. Arshad et al. [29] introduced horizontal fairness concerns behavior into the dual-channel closed supply chain based on manufacturer monopoly, analyzing the influence of decision makers' fairness concerns on the choice of recycling strategy, and the optimal decision of members. Zhang et al. [30] focused on investigating the effects of consumer environmental awareness (CEA) and retailer fairness on environmental quality and green product prices by comparing three different decision scenarios. Xiao et al. [31] investigated the optimal return control problem in a closed-loop supply chain with manufacturers leading and studied the effects of stochastic return disturbance and fairness concerns. Ma et al. [32] introduced marketing effort and fairness concern into CLSCs and examined how retailer fairness concerns affect optimal marketing efforts, recovery rates, and supply chain performance. Zhang et al. [33] based their work on two subsidies provided

by the government: the fixed subsidy and discount subsidy, by constructing a two-echelon supply chain to study the impact of consumer environmental awareness (CEA), retailers' fairness concerns, and government subsidies on members' decisions and compared the advantages of subsidy strategies. This article is similar to the above literatures, and we still use the manufacturer leading model to study the impact of retailer fairness concerns. Kirshner et al. [34] explored the influence of reference effects on supply chains, including internal and external references. Liu et al. [35] investigated the impacts of distributional fairness concerns and peer-induced fairness concerns by comparing the different models in which two FLSPs exhibit four combinations of fairness concerns. In addition, they further proposed incentive contracts to optimize decision-making during order allocation. In terms of fairness concerns, most scholars focus on the influences of the fairness concerns of decision-makers on channel conflicts and pricing, such as in [6,28,31,32,34,35]; only in [29] is the dual-channel discussed when considering fairness. In [30,33], the green manufacturing of products is discussed, and [33] also studied government subsidies. Thus, this paper makes up for that research gap.

From the above literature review, it can be concluded that previous studies have only considered the situation under one or two conditions, such as the fairness concerns of members, channel competition, and a product's green degree. Few studies consider the dual-channel green supply chain operation under members' decision-making behavior. Thus, this paper fills the gap, contributing to an exploration of the effect of retailer fairness concerns on a manufacturer's green manufacturing strategy choices under the dual-channel context. We also analyzed the effect of a product's green degree, government subsidies, and channel competition on members' optimal decisions. We summarize the papers that are most related to our research in Table 1.

**Table 1.** Papers that are most related to our research.

| Author | Dual-Channel | Fairness | Green Product | Subsidies or Tax |
|:---:|:---:|:---:|:---:|:---:|
| Du et al. [14] | | ✓ | ✓ | |
| Chen et al. [16] | | | ✓ | ✓ |
| Zhou et al. [17] | | | ✓ | ✓ |
| Ma et al. [19] | ✓ | | ✓ | |
| Zhang et al. [22] | ✓ | | | ✓ |
| Wang et al. [23] | ✓ | | ✓ | |
| Li et al. [25] | ✓ | ✓ | | |
| Arshad et al. [29] | ✓ | ✓ | | |
| Zhang et al. [30] | | ✓ | ✓ | |
| Zhang et al. [33] | | ✓ | ✓ | ✓ |
| This paper | ✓ | ✓ | ✓ | ✓ |

## 3. Problem Formulation and Notations

This study considers a green dual-channel supply chain system consisting of a single manufacturer and a single retailer, as shown in Figure 1. An assumption is made that the manufacturer produces a single ordinary product before adopting green manufacturing. However, the manufacturer produces a single green product after adopting green manufacturing, as expressed by green degree, but the functional properties of the two products are the same. When the manufacturer adopts green manufacturing, they choose to produce a green product. Similar to Zhu et al. [11], that can be expressed by the green degree, they then place these products into the market via traditional retail channels and their online direct online channel, where the green degree is expressed as $g$, $g \in [0,1]$. However, the manufacturer needs to pay extra costs to produce green products when implementing green manufacturing. Thus, their marginal costs are related to the unit variable costs of products and the green degree of products. The production cost of ordinary products is $c$, and the production cost of green products becomes $c_G = (1+g)c$. The dual-channel supply chain system and the symbols involved are illustrated in Figure 1 and Table 2.

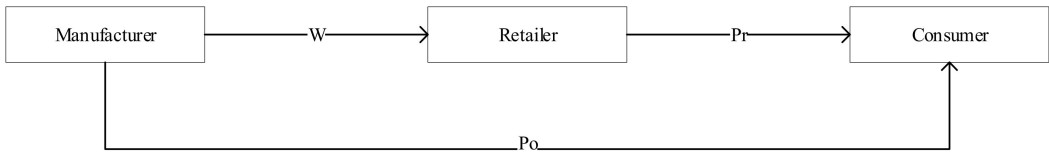

**Figure 1.** Green Dual-Channel Supply Chain System.

**Table 2.** Symbol Description.

| Symbol Name | Symbolic Meaning |
|---|---|
| $p_r^j, p_o^j, w^j$ | Under different modes ($j = NM, GM$), the traditional retail channel price, direct online channel price and wholesale price. |
| $c, c_G$ | Manufacturer's production costs of ordinary products and green products, in which the cost of green products is affected by the green degree of products, expressed as follows: $c_G = (1 + g)c$ |
| $D_r^j, D_o^j$ | The demand of the traditional retail channel and direct online channel under different modes. |
| $Q$ | The overall scale of potential market demand. |
| $\mu$ | The market share of the traditional retail channel; $1 - \mu$ is the market share of the direct online channel, and $0 < \mu < 1$. |
| $\alpha$ | The cross-price sensitivity coefficient of consumers to the traditional retail channel and direct online channel. |
| g | The degree of green products; $0 < g < 1$. |
| $k, k_o$ | The green degree sensitivity of the traditional retail and direct online channel, reflecting the scale of potential additional demand generated by a product's green degree for the traditional retail channel and direct online channel. |
| s | Government subsidies for green products purchased by consumers, meeting the requirement of $s < c < c_G$ |
| $\lambda, \hat{\lambda}$ | The former expresses the retailer's fairness concern coefficient. For the analysis of the model, it will be transformed without changing the results to meet the requirement of $\hat{\lambda} = \lambda/(1 + \lambda)$. |
| $\pi_i^j$ | Under different modes, the profit of each member is represented by the manufacturer, the retailer and the total supply chain system by the subscript $i = m, r, sc$ |
| $U_i^j$ | The utility of each member in different modes; the superscript and the superscript are the same. |

Market demand is affected by consumers' preferences for green products, product prices, and a product's green degree when implementing green manufacturing. In addition, a number of policies for private subsidies for the purchase of new energy vehicles have been introduced in China. For example, if consumers buy purely electric vehicles such as SAIC Roewe ERX5, according to the latest subsidy policy on 26 March 2019, the total amount of actual subsidies available to consumers is 14,400 yuan. Thus, when the government subsidizes consumers to buy green products, this will also affect market demand. In accordance with [26,36–38], this study sets the market demand as a linear structure of product price and the green sensitivity coefficient. The demand functions of traditional retail and online direct online channels for green products are as follows:

$$D_r = \mu Q - (p_r - s) + \alpha(p_o - s) + kg; D_o = (1 - \mu)Q - (p_o - s) + \alpha(p_r - s) + k_o g.$$

$Q$ is the potential market demand and is sufficiently large. $\mu$ is the market share of the traditional channel, thereby reflecting the absolute difference of the consumer demand in the traditional channel.

$1 - \mu$ is the market share of the direct online channel for the manufacturer. $\alpha$ is the cross-price sensitivity coefficient of consumers to the traditional retail and direct online channel, which is also the competition intensity between channels, similar to Yang et al. [26], and expresses the green degree sensitivity of the traditional retail and direct online channel. $k > k_0$ is set because consumers can experience green products better in the traditional retail channel. $kg$ is the market demand of a product's green degree for the traditional retail channel, and $k_o g$ is the market demand of a product's green degree for direct online channel. The current research introduces profit differences into the utility function of decision makers to characterize fairness concerns. Bolton [39] first investigated fairness concerns, believing that positive and negative benefits of fairness concerns would cause losses to their utility. Loch et al. [40] put forward a relatively concise utility form of fairness concern: $U_i(\pi_i) = \pi_i + \theta\pi_j$.

Retail decision-makers with fairness concerns regard the manufacturer's profit as their reference point. The coefficient of fairness concerns is introduced to describe the effect of the fairness concern brought about by the difference of the main body's profit for this reason. When the retailer's profit is greater than the manufacturer's, its utility will increase; otherwise, its utility will decrease. Similar to Du et al. [41], the utility function of fairness concern can be expressed as follows: $U_i(\pi_i) = \pi_i - \lambda(\pi_j - \pi_i)$, in which $\lambda(0 < \lambda < 1)$ represents the retailer's fairness concern coefficient. Note $U_I \equiv U_i/(1+\lambda)$, $\hat{\lambda} = \lambda/(1+\lambda)$ to facilitate the analysis of subsequent models. Only dimension change is involved, and the result is unchanged under this kind of change. The result can still be used as a measure of the decision maker's utility.

Some of the following assumptions need to be supplemented to more fully explain the model in this article:

(1) When a manufacturer implements green manufacturing, the manufacturer produces green products expressed in terms of their green degree. At this time, in order to stimulate consumers to buy green products, the government subsidizes consumers who purchase green products. Referring to Zhang et al. [33], we assume the government subsidies are expressed by $s$, which needs to be met $s < c < c_G$. Because the subsidies should not be greater than the cost of ordinary products manufactured by the manufacturer.

(2) The manufacturer decides whether green manufacturing is implemented or not. The model is the Stackelberg model. The sequence of decision making is as follows: first, the manufacturer makes a decision regarding its direct online price $p_o$ and wholesale price $w$. Then, the retailer makes a decision regarding the price $p_r$ of the traditional retail channel according to the wholesale price given by the manufacturer and the direct online channel price.

Based on the assumptions, the profit function of the manufacturer after adopting green manufacturing can be obtained as follows:

$\pi_m = (p_o - c_G)D_o + (w - c_G)D_r$, that is

$$\pi_m = (p_o - c_G)[(1 - \mu)Q + k_o g - (p_o - s) + \alpha(p_r - s)] + (w - c_G)[\mu Q + kg - (p_r - s) + \alpha(p_o - s)]. \quad (1)$$

The retailer's profit function is

$$\pi_r = (p_r - w)D_r = (p_r - w)[\mu Q + kg - (p_r - s) + \alpha(p_o - s)]. \quad (2)$$

Two models of a manufacturer's green dual-channel supply chain are established under the circumstance of a retailer's fairness concerns to analyze the optimal decision of the manufacturer to implement green manufacturing, and the impacts of a retailer's fairness concern son manufacturer's decision making of implementing a green supply chain, as follows: (1) the Stackelberg model (*NM*) for the manufacturer not to implement green manufacturing; and (2) the Stackelberg model (*GM*) for the manufacturer to implement green manufacturing.

## 4. Model Establishment and Analysis

*4.1. Stackelberg Model without Green Manufacturing (NM)*

A manufacturer does not produce green products without adopting green manufacturing. A manufacturer sells ordinary products via the direct online channel and traditional retail channel. The manufacturer occupies a dominant position in the market. First, the manufacturer makes decisions on the wholesale price $w$ and direct online price $p_o$. Then, the retailer with fairness concerns makes a decision regarding the retail price $p_r$ according to the wholesale price and direct online price given by the manufacturer. At this time, the demands of traditional retail and direct online channels are as follows:

$$D_r = \mu Q + kg - p_r + \alpha p_o; \; D_o = (1 - \mu)Q + k_o g - p_o + \alpha p_r.$$

Furthermore, the profits of the manufacturer and retailer are as follows:

$$\pi_m^{NM} = (p_o - c)[(1 - \mu)Q - p_o + \alpha p_r] + (w - c)(\mu Q - p_r + \alpha p_o)$$

$$\pi_r^{NM} = (p_r - w)(\mu Q - p_r + \alpha p_o).$$

When the retailer has fairness concerns, the retailer's utility function is

$$\text{Max}U_r^{NM} = \pi_r^{NM} - \hat{\lambda}\pi_m^{NM} = (p_r - w)(\mu Q - p_r + \alpha p_o) - \hat{\lambda}(p_o - c)[(1 - \mu)Q - p_o + \alpha p_r] - \hat{\lambda}(w - c)(\mu Q - p_r + \alpha p_o). \quad (3)$$

The utility function of the manufacturer is the same as its profit function, that is

$$\text{Max}U_m^{NM} = \pi_m^{NM} = (p_o - c)[(1 - \mu)Q - p_o + \alpha p_r] + (w - c)(\mu Q - p_r + \alpha p_o). \quad (4)$$

This game model can be solved by using backwards induction. From Equations (3) and (4), we can derive Lemma 1.

**Lemma 1.** *Under this model, when* $-2 + (1 - \hat{\lambda})\alpha^2 < 0$*, the optimal equilibrium decision of the retailer and manufacturer is*

$$p_o^{NM*} = \frac{(\alpha\mu - \mu + 1)Q}{2(1 - \alpha^2)} + \frac{c}{2}, w^{NM*} = \frac{(1 + \hat{\lambda}\alpha^2)\mu Q + 2\alpha(1 - \mu)Q}{2(1 - \alpha^2)(1 + \hat{\lambda})} + \frac{(1 + 2\hat{\lambda} - \alpha\hat{\lambda})c}{2(1 + \hat{\lambda})},$$

$$p_r^{NM*} = \frac{(3 - \alpha^2)\mu Q + 2\alpha(1 - \mu)Q}{4(1 - \alpha^2)} + \frac{(1 + \alpha)c}{4}.$$

*All proofs are given in Appendix A.*

From Lemma 1, we can see that the traditional retail price, direct online price, and wholesale price are all affected by the channel market share, channel cross price sensitivity coefficient, potential market demand scale, and production cost. However, only the wholesale price is affected by the retailer's fairness concern, whereas traditional retail and direct online selling prices are not affected, which is directly reflected in the retailer's bargaining power with the manufacturer under the influence of fairness concerns.

Further, we can obtain the demand of each channel at this time as follows:

$$D_r^{NM*} = \frac{\mu Q - (1 - \alpha)c}{4}; D_o^{NM*} = \frac{\mu Q\alpha + 2(1 - \mu)Q - (1 - \alpha)(2 + \alpha)c}{4}.$$

*4.2. Stackelberg Model with Green Manufacturing (GM)*

In this model, the manufacturer implements green manufacturing, produces green products that can be expressed by their green degree, and sells them simultaneously through direct online and

traditional retail channels. The manufacturer still dominates as market leader, whereas the retailer with fairness concerns is a follower. The sequence of decision making is that the manufacturer first decides its direct selling price $p_o$ and wholesale price $w$. Then, the retailer makes a decision on the retail price $p_r$ according to the manufacturer's decision. When the government subsidizes the green product consumers per unit $s$, the actual prices of the products purchased by consumers are $p_o - s$ and $p_r - s$. After adopting green manufacturing and being subsidized by the government, the demand of direct and retail channels become the following:

$$D_r = \mu Q + kg - (p_r - s) + \alpha(p_o - s); D_o = (1 - \mu)Q + k_o g - (p_o - s) + \alpha(p_r - s).$$

At this time, the profit functions of manufacturer and retailer are

$$\pi_m^{GM} = (p_o - c_G)[(1 - \mu)Q + k_o g - (p_o - s) + \alpha(p_r - s)] + (w - c_G)[\mu Q + kg - (p_r - s) + \alpha(p_o - s)]$$

$$\pi_r^{GM} = (p_r - w)[\mu Q + kg - (p_r - s) + \alpha(p_o - s)].$$

Then, the decision utility function of retailer and manufacturer can be obtained as follows:

$$\text{MaxU}_r^{GM} = (p_r - w)[\mu Q + kg - (p_r - s) + \alpha(p_o - s)] - \hat{\lambda}(p_o - c_G)[(1 - \mu)Q + k_o g - (p_o - s) + \alpha(p_r - s)]$$

$$-\hat{\lambda}(w - c_G)[\mu Q + kg - (p_r - s) + \alpha(p_o - s)] \tag{5}$$

$$\text{MaxU}_m^{GM} = \pi_m^{GM} = (p_o - c_G)[(1 - \mu)Q + k_o g - (p_o - s) + \alpha(p_r - s)] + (w - c_G)[\mu Q + kg - (p_r - s) + \alpha(p_o - s)] \tag{6}$$

As shown above, we obtain Lemma 2. The backwards induction is still used, so the proof process of Lemma 2 is similar to Lemma 1, and so we omit it here.

**Lemma 2.** *Under the GM model, when the* $-2 + \alpha^2(1 - \hat{\lambda}) < 0$ *condition is satisfied, the retailer and the manufacturer balance the optimal decision-making position:*

$$p_o^{GM*} = \frac{2[(1 - \mu)Q + k_o g + s + c_G] + (\mu Q + kg + c_G - s)\alpha + [(2\hat{\lambda} - 1)c_G - s]\alpha^2}{2[2 - \alpha^2(1 - \hat{\lambda})]}$$

$$w^{GM*} = \frac{\mu Q + kg + (1 + \alpha + 2\hat{\lambda})c_G + (1 - \alpha)s}{2(1 + \hat{\lambda})}$$

$$p_r^{GM*} = \frac{3(\mu Q + kg + s) + c_G + (1 - \hat{\lambda})[(1 - \mu)Q + k_o g]\alpha + (2 + \hat{\lambda})(c_G + s)\alpha - (1 - \hat{\lambda})[(\mu Q + kg + 2s)\alpha^2 + (c_G - s)\alpha^3]}{2[2 - \alpha^2(1 - \hat{\lambda})]}.$$

From Lemma 2, the traditional retail price, direct online price, and wholesale price are all affected by the potential market demand scale, channel market share, channel cross price sensitivity coefficient, and product cost, as with Lemma 1. In addition, the prices are also affected by the green degree of green products, the scale coefficient of potential additional demand generated by the green degree of products for various channels, and the government subsidies to consumers. Interestingly, after adopting green manufacturing, a retailer's fairness concerns impact wholesale prices and the prices of each channel.

At this time, we can determine the demands of each channel, as follows:

$$D_r^{NM*} = \frac{(1 + \hat{\lambda}\alpha^2)(\mu Q + kg) + \alpha(1 + \hat{\lambda})[(1 - \mu)Q + k_o g] - (1 + \alpha - \alpha^2 - \hat{\lambda}\alpha^3)(c_G - s)}{2[2 - \alpha^2(1 - \hat{\lambda})]}$$

$$D_o^{NM*} = \frac{(\mu Q + kg)\alpha + (1 - \mu)Q + k_o g - (1 - \alpha^2)(c_G - s)}{2}.$$

**Proposition 1.** *Under green manufacturing:*

(1)　$\partial p_o^{GM*}/\partial g > 0, \partial w^{GM*}/\partial g > 0, \partial p_r^{GM*}/\partial g > 0$;
(2)　$\partial D_r^{GM*}/\partial g > 0, \partial D_o^{GM*}/\partial g > 0$.

Proposition 1 demonstrates that the wholesale price and channel price of the product are positively related to the green degree of the product. The main reason for this is that after adopting green manufacturing, the cost of products will increase. Thus, the manufacturer will increase wholesale prices and direct online prices to ensure profits, whereas the retailer will increase the retail price to a certain extent to avoid harming their marginal revenue. We can note that despite competition among channels occurring after adopting green manufacturing. However, the expansion effect of green products will offset the side effects of channel competition with the improvement of the product's green degree, and consumers are willing to pay a certain extra fee to buy green products, so the demand of products under dual channels will increase.

For proof, see Appendix B.

**Proposition 2.** *After implementing green manufacturing:*

(1)　$\partial p_o^{GM*}/\partial s > 0, \partial w^{GM*}/\partial s > 0, \partial p_r^{GM*}/\partial s > 0$
(2)　$\partial D_r^{GM*}/\partial s > 0, \partial D_o^{GM*}/\partial s > 0$.

Proposition 2 presents the same conclusion as [33]: after a manufacturer adopts green manufacturing, as government subsidies increase, the channel price of products will increase, and the channel demand will also advance. This shows that the government subsidies for consumers who buy green products have stimulated their demand. With increasing demand, the manufacturer increases prices to a certain extent to obtain more profits, whereas the retailer increases their price accordingly to ensure their channel revenue and product competitiveness. In addition, we find that although the government subsidizes consumers, the manufacturer will increase profits by increasing channel prices, so the manufacturer will actually enjoy the benefits of government subsidies.

Certification is shown in Appendix B.

*4.3. Contrastive Analysis of NM and GM Models*

Through further analysis and discussion of the theoretical models obtained, this section hopes to analyze and explain the internal reasons behind the green dual-channel supply channel. Besides, we provide several pieces of management inspiration.

**Corollary 1.** *When green manufacturing is not adopted, the direct online price and retail price of products are not affected by the characteristics of a retailer's fairness concerns, whereas the wholesale price will decrease with the increase of fairness concerns. On the contrary, the impact of a retailer's fairness concerns will expand and the wholesale price, direct online price, and retail price of products will decrease with the increase of fairness concerns.*
*Certification is shown in Appendix C.*

Corollary 1 shows that whether green manufacturing is adopted or not, the characteristics of a retailer's fairness concerns have a direct impact on the wholesale price offered by the manufacturer. The reason for this impact is that the bargaining power between retailer and manufacturer increases with the improvement of retailer's fairness concerns. The retailer will force the manufacturer to reduce the wholesale price to a certain extent to ensure that their unit profit is not damaged and to ensure fairness. When this strategy is not adopted, the retail price is unaffected, mainly because the retailer wants to ensure that he or she can compete with the direct online channel, thereby maintaining the same trend as the manufacturer's online direct online price. In contrast, when the manufacturer adopts green manufacturing, combining Propositions 1 and 2, the green degree of the product and

government subsidies all have a positive effect on price and demand. Therefore, facing the increase of the manufacturer's profit, the retailer's sense of fairness is intensified, and the negative utility will increase and expand. Thus, at this time, the wholesale price and each channel price are affected and decrease with the increase of fairness.

**Proposition 3.** *Before and after the implementation of green manufacturing, the wholesale price, product channel price, and demand are related as follows:*

(1)　$p_o^{NM*} > p_o^{GM*}, p_r^{NM*} > p_r^{GM*}, w^{NM*} > w^{GM*}$
(2)　$D_r^{NM*} < D_r^{GM*}, D_o^{NM*} < D_o^{GM*}.$

*The certification is shown in Appendix B.*

As shown by Propositions 1 and 2, after the implementation of green manufacturing, the wholesale, retail, and direct channel prices of products will increase with increasing government subsidies and the green degree of products. Similarly, the demand of all channels will increase. Combined with Proposition 3, the demand of each channel is higher than without the implementation of green manufacturing because of the influence of government subsidies and consumers' willingness to pay extra for green products. Simultaneously, the manufacturer indirectly enjoys government subsidies, which offset the additional cost of implementing green manufacturing. Then, the manufacturer further reduces the wholesale price. Thus, the direct selling and retail prices obtain a reduction space, and the price is reduced to a certain extent. In reality, battery electric vehicles (BEVs) are cheaper than the corresponding fuel vehicles (FVs); for example, in 2016, the manufacturer's suggested retail price of a Volkswagens e-Golf was $32,157, while the basic diesel Golf was $33,226 [42]. From this point of view, the green manufacturing strategy adopted by the manufacturer is beneficial to the manufacturer and retailer and is also a powerful factor affecting consumers' purchases of environmental and green products at low prices.

The following corollary analyzes the impact of retailer fairness concerns on product prices before and after adopting green manufacturing strategy.

**Corollary 2.** *Before and after adopting green manufacturing, the impacts of a retailer's fairness concerns on product channel prices and wholesale prices are as follows:*

$$\partial(p_o^{NM*} - p_o^{GM*})/\partial\hat{\lambda} > 0;\ \partial(p_r^{NM*} - p_r^{GM*})/\partial\hat{\lambda} > 0;\ \partial(w^{NM*} - w^{GM*})/\partial\hat{\lambda} > 0.$$

*The certification is shown in Appendix C.*

Corollary 2 shows that the wholesale and channel prices are lower than those without green manufacturing, and the price gap will increase with the increase of retailers' fairness concerns. Combining Corollary 1, under the adoption of green manufacturing, the impact of the fairness concerns of retailers will expand, which will further affect channel prices. Thus, with the increase of fairness, retailers will get more profit by lowering the channel price to get more channel demand and ensuring that they are treated fairly in the supply chain. At the same time, in order to ensure the competitiveness of direct online channel products, the manufacturer will also reduce prices accordingly, resulting in an increase in the price gap under the influence of fairness concerns. From the consumers' point of view, the retailer's consideration of the green dual-channel supply chain is beneficial under the characteristics of fairness concerns. Therefore, while benefitting from government subsidies, consumers can further enjoy low-cost green products.

**Corollary 3.** *When the manufacturer does not implement the green manufacturing strategy, the retailer's fairness will not affect the demand of each channel. The channel demands are only related to the market scale, market share, product cost, and channel price sensitivity coefficient. After adopting the green manufacturing*

*strategy, the direct online channel still has nothing to do with the retailer's fairness concerns, but the retail channel's demand will rise with increasing fairness concerns.*

*The certification is shown in Appendix C.*

## 5. Numerical Simulation Analysis

Based on the hypothesis, the market basic capacity and potential demand $Q = 200$, and the market share of traditional retail channels $\mu = 0.6$. In addition, $k = 40$, $k_o = 20$, $c = 10$, $c_G = 15$, $\alpha = 0.5$, and $s = 2$. Matlab 2016b was used for numerical simulation to verify the conclusions drawn from the analysis and to analyze the choice of the green manufacturing strategy of the dual-channel supply chain under the influence of retailers' fairness concerns.

### 5.1. Sensitivity Analysis of Retailers' Fairness Concerns

This section focuses on the impact of retailer's fairness concerns on members' decisions, profit and utility before and after a manufacturer adopts a green manufacturing strategy. Taking a retailer's fairness concern $\hat{\lambda} \in [0,1]$, this study compares the wholesale price, direct online price, and traditional retail price, as well as the demand of each channel and profit utility of members.

As can be seen from Figure 2a–c, when the manufacturer does not implement a green manufacturing strategy, the direct online direct selling price and retail price are not affected by the retailer's fairness concern $\hat{\lambda}$ and remain unchanged. The wholesale price of traditional retail channels will be affected and will decrease with the increase of $\hat{\lambda}$, and the demand of direct online and traditional retail channels will remain unchanged. When the manufacturer implements a green manufacturing strategy, the impact of the retailer's fairness concerns will widen, as will online direct selling and traditional retail prices, whereas wholesale prices are still affected by their fairness concern. The trend of decrease is maintained when increasing. This also verifies the conclusion of Corollary 1. Simultaneously, observing Figure 2c, a retailer's fairness concern only has an impact on traditional retail channel demand when a manufacturer adopts a green manufacturing strategy, and it increases as fairness concerns increase. At this time, the direct online channel demand and the demand of each channel in the case of the manufacturer not adopting green manufacturing remain unchanged, and all are not affected by a retailer's fairness concerns. Moreover, the demand of each channel under the strategy of green manufacturing are higher than that without the strategy of green manufacturing. Thus, when a retailer's fairness concerns increase, the manufacturer can reduce the wholesale price and direct online price to increase profits and utility.

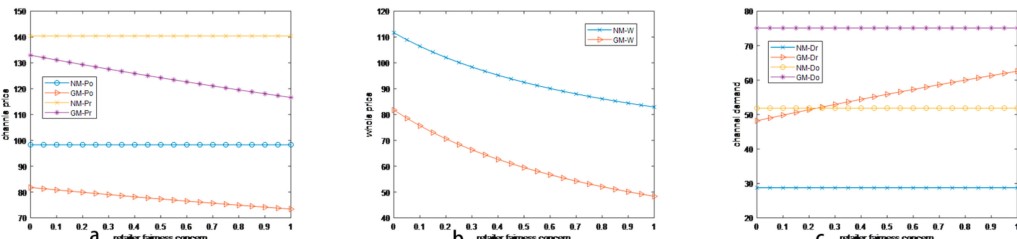

**Figure 2.** The impact of $\hat{\lambda}$ on (**a**) channel price, (**b**) wholesale price, and (**c**) demand.

As shown in Figure 3, regardless of whether the manufacturer adopts green manufacturing or not, the profit of the manufacturer maintains a declining trend with the improvement of the retailer's fairness concern level, whereas the opposite outcome is achieved for the retailer. The higher the level of fairness concerns, the higher the retailer's profit will be. When the manufacturer adopts green manufacturing, the retailer's profit is always higher than when the manufacturer does not adopt green manufacturing, In addition, we found that when the retailer has a sufficiently high fairness level

(namely $\hat{\lambda} > 0.75$), the manufacturer will not adopt green manufacturing because the manufacturer's profit is lower.

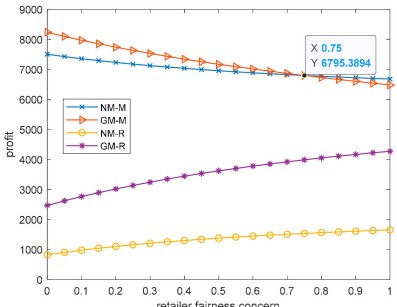

**Figure 3.** The effect of $\hat{\lambda}$ on the profits and utility.

As can be observed in Figure 4, the impacts of retailer's fairness concerns is the same because the utility of the manufacturer is equal to the profit of the manufacturer. Thus, the manufacturer's strategy choice is consistent with the above analysis. In addition, regardless of whether the manufacturer adopts a green manufacturing strategy or not, the retailer's utility decreases with the improvement of the fairness concerns level. At the same time, under green manufacturing, a retailer's utility is higher than without green manufacturing. The retailer will benefit from the green manufacturing strategy, and the retailer will choose low fairness concerns to ensure relatively high utility.

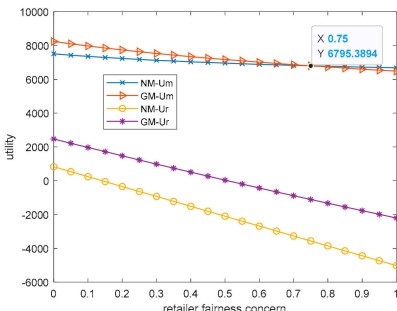

**Figure 4.** The effect of $\hat{\lambda}$ on the supply chain members' utility.

Combining Figures 2–4 and looking at Figure 5, under green manufacturing, the total profit of the system is higher than that without green manufacturing, which shows that the green manufacturing strategy promotes the total profit of the system. Otherwise, the total profit of the system remains unchanged with the improvement of the retailer's fairness concerns without green manufacturing, whereas under green manufacturing, the total profit of the system will first rise, then decrease with the increase of the retailer's fairness concerns. This is because under green manufacturing, when the retailer's fairness concern is low, the decrease of the manufacturer's profit is less than the increase of retailer's profit, and so the total profit will rise. When the retailer's fairness concern is high, the bargaining power of the retailer increases, which leads to the decrease of the manufacturer's profit more than the increase of retailer's profit. Therefore, the total profit shows a downward trend. By observing the overall utility of the system, regardless of whether the manufacturer adopts green manufacturing, the system utility is negatively to the retailer's fairness concerns. This shows that the retailer tends to sacrifice profits to gain a sense of fairness in the channel, which will also harm the supply chain system. Moreover, from the perspective of the overall supply chain system, the green manufacturing strategy is preferred and the retailer's fairness maintained at a low level.

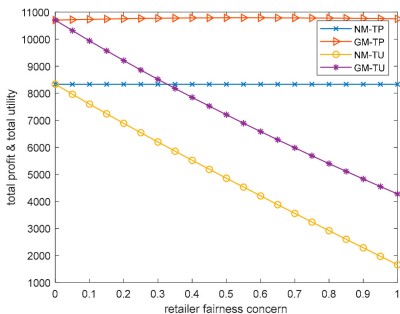

**Figure 5.** The effect of $\hat{\lambda}$ on system profit and utility.

### 5.2. Sensitivity Analysis on g, s, k and $k_o$

The green degree of the manufacturer's products is analyzed and studied, and the relevant parameters are still the same. Here, the green degree of the manufacturer's products is $g \in [0,1]$, $\hat{\lambda} = 0.4$.

As can be seen from Figure 6a,b, the wholesale price, price, and demand of each channel will increase with the green degree of the product. This result proves Proposition 1 and shows that the manufacturer's adoption of green manufacturing can effectively stimulate the market and increase demand. Observing Figure 6c, a product's green degree has a positive effect on members' profit and utility. This reflects that the manufacturer adopting green manufacturing to produce high green-degree products is beneficial to supply-chain members.

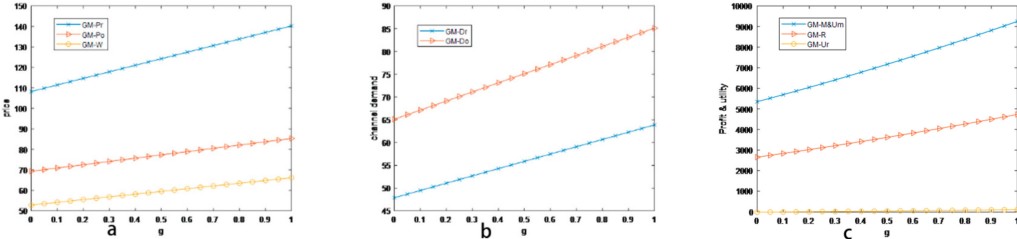

**Figure 6.** The impact of *g* on (**a**) price, (**b**) demand, and (**c**) members' profit and utility.

It can be seen from Figure 7a–c that the increase of a product's green degree and government subsidies to consumers all contribute to the improvement of profit and the utility of manufacturer and retailer. This is mainly due to an increase in the product's green degree and government subsidies, which greatly increases demand. This can also be seen in Propositions 1 and 2. In addition, combining Figures 2 and 6, it was found that the increase of a product's green degree and government subsidies had a counter effect which can alleviate the negative effects of a retailer's fairness concerns to a certain extent.

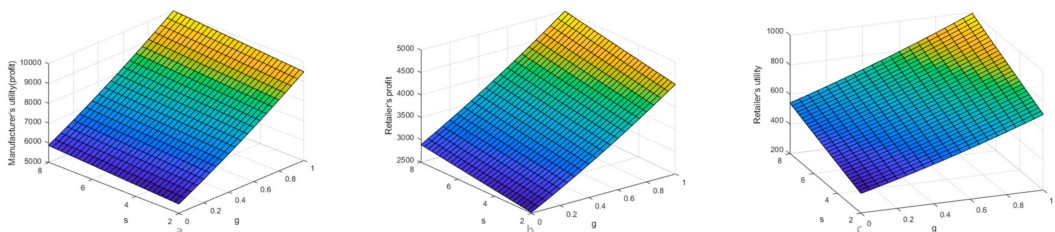

**Figure 7.** The effect of *g* and *s* on (**a**) manufacturer's profit and utility, (**b**) retailer's profit, and (**c**) retailer's utility.

Figure 8a–c reflects the impact of the green sensitivity of traditional retail and direct online channels on the profit and utility of supply chain members. As can be seen from a and b, the green

degree sensitivity of traditional retail and direct online channels has a positive effect on the profit and utility of the manufacturer and the retailer's profit. Observing c, as *k* improves, the retailer's utility presents a decreasing trend. This is because when the retailer has increased fairness concerns, the manufacturer gains more profit through direct online channels, and the retailer feels unfairly treated and their utility decreases.

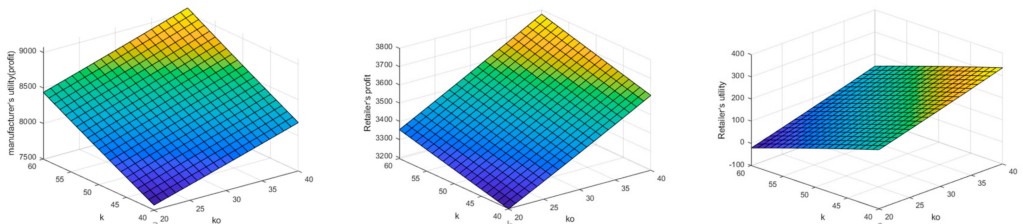

**Figure 8.** The effect of *k* and $k_o$ on (**a**) the manufacturer's utility and profit, (**b**) retailer's profit, and (**c**) retailer's profit and utility.

### 5.3. Sensitivity Analysis on $\alpha$ and $\mu$

The cross-price sensitivity coefficient and the market share of the traditional retail channel is analyzed, and the relevant parameters are still the same as above. Here, the cross-price sensitivity coefficient and the market share of the traditional retail channel are $\alpha \in [0,1]$ and $\mu \in [0,1]$, respectively.

It can be seen from Figure 9a–c that the increase in channel crossover coefficient indicates that channel competition is intensified. When a retailer has fairness concerns, the wholesale price, price and demand of each channel all increase with channel crossover coefficient. In addition, it can be found that under green manufacturing, each channel price of green products are lower than that without green manufacturing. The reason behind this is mainly due to the retailer's fairness, the green degree of products and government subsidies. This is also consistent with real life, such as in battery electric vehicles (BEVs) which are cheaper than the corresponding fuel vehicles (FVs); in 2016, the manufacturer's suggested retail price of a Volkswagens e-Golf was $32,157, while the basic diesel Golf was $33,226 [42].

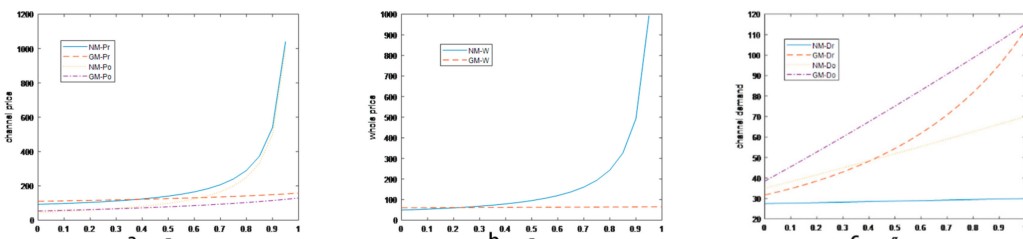

**Figure 9.** The effect $\alpha$ on (**a**) channle price, (**b**) wholesale price, and (**c**) demand.

To clearly analyze the effect of market share, we show the result from the perspective of $\mu$ in Figure 10. We found that as the share of the traditional retail channel market share $\mu$ increases, both the wholesale price and retail price will increase, while direct channels will be reversed. The change in demand is also the same. This shows that the increase in the proportion of the traditional channel market share is beneficial to the traditional retail channel.

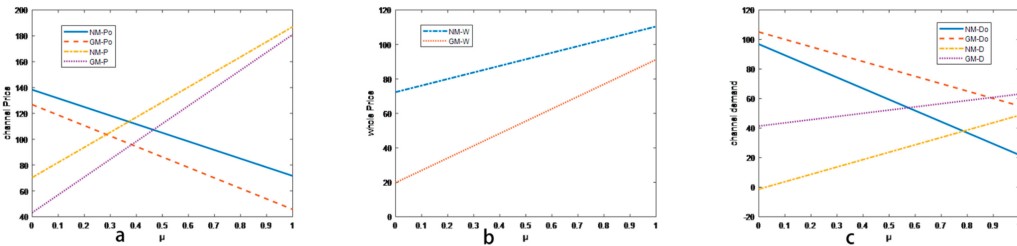

**Figure 10.** The effect $\mu$ on (**a**) channel price, (**b**) wholesale price, and (**c**) demand.

Figure 11 reflects that when the retailer has fairness concerns, $\alpha$ and $\mu$ have an impact on the adoption choice of green manufacturing under a dual-channel system. Figure 11a shows the manufacturer's strategy preferences, and Figure 11b shows the retailer's strategy preferences. It can be seen that when the channel price crossover coefficient is increased—that is, the competition between channels is intensified—both the manufacturer and retailer tend toward the traditional dual channel model. When $\alpha$ is small ($\alpha$ is approximately less than 0.45), the manufacturer will tend to adopt green manufacturing; while $\alpha$ and $\mu$ are small ($\alpha$ is approximately less than 0.3 and $\mu$ is approximately less than 0.4), the retailer prefers the manufacturer to adopt green manufacturing. This reflects the fact that when the retailer has fairness concerns, it is not always beneficial for supply chain members to adopt green manufacturing under a dual-channel supply chain. Therefore, whether the manufacturer adopts green manufacturing should be based on the retailer's fairness concerns, the product's green degree, government subsidies, channel competition and market share.

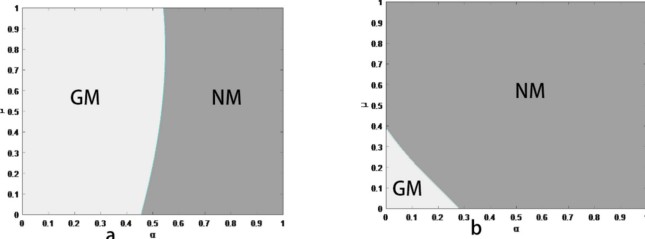

**Figure 11.** Impact of $\alpha$ and $\mu$ on (**a**) manufacturer's green manufacturing strategy preferences, and (**b**) retailer's green manufacturing strategy preferences.

## 6. Conclusions

Under the background of a dual-channel system, this paper constructs a green dual-channel supply chain model to analyze whether a manufacturer adopts a green manufacturing strategy when the retailer has fairness concerns and comprehensively considers the impact of a product's green degree, government subsidies, channel competition, and market share on the members' optimal decision-making. Two scenarios were demonstrated and comparatively analyzed.

From the comparisons and discussion of the decisions among the two scenarios, we obtained interesting managerial findings: (1) Under a green manufacturing strategy, the impact of a retailer's fairness concerns will expand and has a negative effect on the channel price. In particular, the manufacturer will not adopt green manufacturing when a retailer's fairness concern is sufficiently high ($\hat{\lambda} > 0.75$) or the channel price crossover coefficient is relatively low ($\alpha$ is approximately less than 0.45). While $\alpha$ and $\mu$ are small ($\alpha$ is approximately less than 0.3 and $\mu$ is approximately less than 0.4), the retailer prefers the manufacturer to adopt green manufacturing. Thus, from the perspective of supply chain members, green manufacturing is not always beneficial. From the overall utility of the system, the supply chain is more effective under green manufacturing and tends to keep the retailer's fairness concerns at a low level. (2) Under green manufacturing, the product's green degree and government subsidies have a positive effect on the channel price, demand, and the members' profit and utility. Additionally, government subsidies can alleviate the negative effects of a retailer's fairness concerns to

a certain extent. (3) Comparing the two scenarios (*NM & GM*), we found that the channel price of the *GM* model is lower than the *NM* model. This differs from previous studies, but is also consistent with reality, such as for battery electric vehicles (BEVs), which are cheaper than the corresponding fuel vehicles (FVs); for example, in 2016, the manufacturer's suggested retail price of a Volkswagens e-Golf was \$32,157, while the basic diesel Golf was \$33,226 [42].

This study does have limitations. From the research results, due to the existence of the preferences of the decision-making members, the loss of the utility of the supply chain benefits. In the future, we need to consider how to effectively supply the supply chain through cooperation between the supply chain and appropriate competition. Moreover, the main model is considered with complete information, but in reality, the information is incomplete. In addition, multiple retailers or manufacturers often simultaneously compete in decision making. This study does not consider such a situation, let alone the impact of their competitive relationship and intensity. At the same time, adopting GSCM often involves numerous aspects, including enterprise operation, product production, and a reverse supply chain. Therefore, subsequent research can include a reverse supply chain and other aspects in the comprehensive research and analysis.

**Author Contributions:** Z.Z. conceived the paper. Z.Z. and H.Z. wrote the paper. X.P. provided guidance on article structure logic. Y.L. provided guidance on article model analysis.

**Funding:** This work was supported by the National Social Science Fund Project (Grant Numbers 15BGL063).

**Conflicts of Interest:** The authors declare no conflict of interest.

## Appendix A

**Proof of Lemma 1.** According to Equation (4), the first derivative of retail price is obtained and is equal to 0. We can derive the response function of retail price:

$$p_r = \frac{\mu Q + (1 - \hat{\lambda})\alpha p_o + (1 + \hat{\lambda})w - (1 - \alpha)\hat{\lambda}c}{2}.$$

Then, this is substituted by Equation (3), and the Hassian matrix of Equation (3) with respect to $w$ and $p_o$, namely

$$|H| = \begin{vmatrix} -2 + (1 - \hat{\lambda})\alpha^2 & \alpha(\hat{\lambda}+1) \\ \alpha(\hat{\lambda}+1) & -\hat{\lambda} - 1 \end{vmatrix} = (-2\hat{\lambda} - 2)\alpha^2 + 2\hat{\lambda} + 2 > 0, \; |H_1| = -2 + (1 - \hat{\lambda})\alpha^2 < 0.$$

Because $|H_1| = -2 + (1 - \hat{\lambda})\alpha^2 < 0$, and so the Heisen matrix is definitely negative. Then, Formula (3) is a joint concave function of $w$ and $p_o$. There exists a unique solution to solve the reciprocal first order of Formula (3) regarding $w$ and $p_o$ separately and make it equal to 0, namely

$$p_o^{NM*} = \frac{(\alpha\mu - \mu + 1)Q}{2(1 - \alpha^2)} + \frac{c}{2}, \; w^{NM*} = \frac{(1 + \hat{\lambda}\alpha^2)\mu Q + 2\alpha(1 - \mu)Q}{2(1 - \alpha^2)(1 + \hat{\lambda})} + \frac{(1 + 2\hat{\lambda} - \alpha\hat{\lambda})c}{2(1 + \hat{\lambda})}.$$

Subsequently, we substituted them into the response function of the retail price to get the optimal retail price as follows:

$$p_r^{NM*} = \frac{(3 - \alpha^2)\mu Q + 2\alpha(1 - \mu)Q}{4(1 - \alpha^2)} + \frac{(1 + \alpha)c}{4}.$$

This completes the proof.　□

## Appendix B

**Proof of Proposition 1.** (1) From the expressions of $p_o^{GM*}$, $w^{GM*}$ and $p_r^{GM*}$, the first derivative of the product's green degree $g$ can be obtained:

$$\frac{\partial p_o^{GM*}}{\partial g} = \frac{(2\alpha^2\hat{\lambda} - \alpha^2 + \alpha + 2)c - \alpha k + 2k_0}{2[2 - \alpha^2(1 - \hat{\lambda})]} > 0, \quad \frac{\partial w^{GM*}}{\partial g} = \frac{(1 + 2\hat{\lambda} + \alpha)c + k}{2(1 + \hat{\lambda})} > 0,$$

$$\frac{\partial p_r^{GM*}}{\partial g} = \frac{[1 - (1 - \hat{\lambda})\alpha^3 + (2 + \hat{\lambda})\alpha]c + [3 - (1 - \hat{\lambda})\alpha^2]k + (1 - \hat{\lambda})k_0\alpha}{2[2 - \alpha^2(1 - \hat{\lambda})]} > 0.$$

Then, we can conclude that the first derivatives of $p_o^{GM*}$, $w^{GM*}$ and $p_r^{GM*}$ with regard to the product green degree $g$ are all greater than zero, and that the retail price, manufacturer's wholesale price and online direct selling price increase with the increase of product greenness.

(2) For $D_r^{GM*}$ and $D_o^{GM*}$, the first derivative of the product's green degree $g$ is obtained:

$$\frac{\partial D_r^{GM*}}{\partial g} = \frac{(1 + \hat{\lambda})k + (1 + \hat{\lambda})k_0}{2[2 - \alpha^2(1 - \hat{\lambda})]} > 0, \quad \frac{\partial D_o^{GM*}}{\partial g} = \frac{k\alpha + k_0}{2} > 0.$$

It can be concluded that, under the green dual-channel supply chain model, the demand of each channel also increases with the increase of the product's green degree $g$.

This completes the proof. □

**Proof of Proposition 2.** (1) From the expressions of $p_o^{GM*}$, $w^{GM*}$ and $p_r^{GM*}$, the first reciprocal of government subsidy $s$ can be obtained:

$$\frac{\partial p_o^{GM*}}{\partial s} = \frac{2 - \alpha - \alpha^2}{2[2 - \alpha^2(1 - \hat{\lambda})]} > 0, \quad \frac{\partial w^{GM*}}{\partial s} = \frac{1 - \alpha}{2(1 + \hat{\lambda})} > 0, \quad \frac{\partial p_r^{GM*}}{\partial s} = \frac{3(1 - \alpha) + (1 + \hat{\lambda}\alpha)(1 - \alpha)^2}{2[2 - \alpha^2(1 - \hat{\lambda})]} > 0.$$

It can be concluded that the first partial derivatives of $p_o^{GM*}$, $w^{GM*}$ and $p_r^{GM*}$ regarding the government subsidy $s$ are all greater than zero; that is, the retail price, manufacturer's wholesale price and online direct selling price increase with the increase of the government subsidy.

(2) For $D_r^{GM*}$ and $D_o^{GM*}$, the first reciprocal of the government subsidy $s$ is obtained:

$$\frac{\partial D_r^{GM*}}{\partial s} = \frac{(1 - \alpha^2)(1 + \hat{\lambda}\alpha)}{2[2 - \alpha^2(1 - \hat{\lambda})]} > 0, \quad \frac{\partial D_o^{GM*}}{\partial s} = \frac{1 - \alpha^2}{2} > 0.$$

This completes the proof. □

**Proof of Proposition 3.**

(1) The price of each channel and wholesale price are respectively compared, namely

$$p_o^{GM*} - p_o^{NM*} = \frac{2[\hat{\lambda}\alpha^2\mu + (1 + \hat{\lambda})(1 - \mu)\alpha + \mu]\alpha Q - 2(1 - \alpha^2)[2 - (1 - \hat{\lambda})\alpha^2 + \alpha(1 + \hat{\lambda}\alpha)]c_G}{4[2 - \alpha^2(1 - \hat{\lambda})](\alpha^2 - 1)}$$

$$\frac{-2(1 - \alpha^2)[(2 - \alpha - \alpha^2)s + (1 - \hat{\lambda})c\alpha^2 + \alpha kg + 2k_0g - 2c]}{4[2 - \alpha^2(1 - \hat{\lambda})](\alpha^2 - 1)} < 0,$$

$$p_r^{GM*} - p_r^{NM*} = \frac{\alpha^2\mu Q[(1 - \hat{\lambda})\alpha^2 + 3 + \hat{\lambda}] + 2\alpha(1 + \hat{\lambda})(1 - \mu)Q + 2kg[\alpha^2(1 + \alpha^2)(1 - \hat{\lambda}) - 3(1 - \alpha^2)]}{4[2 - \alpha^2(1 - \hat{\lambda})](\alpha^2 - 1)}$$

$$\frac{-2\alpha(1-\alpha^2)(1-\hat{\lambda})k_og + (1+\alpha)(1-\alpha^2)[2-\alpha^2(1-\hat{\lambda})]c_G + \alpha^2(1+\alpha)[2+(1-\alpha^2)(1-\hat{\lambda})]c}{4[2-\alpha^2(1-\hat{\lambda})](\alpha^2-1)}$$

$$\frac{+2[\alpha(2-2\hat{\lambda}\alpha+5\alpha+\hat{\lambda})-3(1+\alpha^3)-\alpha^4(1-\hat{\lambda})(2-\alpha)]s}{4[2-\alpha^2(1-\hat{\lambda})](\alpha^2-1)} < 0,$$

$$w^{GM*} - w^{NM*} = \frac{[(1+\hat{\lambda})\mu Q + kg]\alpha^2 + \alpha(1+\hat{\lambda})(1-\mu)Q - k_og - (1-\alpha)(1-\alpha^2)s}{2(1+\hat{\lambda})(\alpha^2-1)}$$

$$\frac{-\alpha(1-\alpha^2)(\hat{\lambda}c+c_G)-(1-\alpha^2)(1+2\hat{\lambda})(c_G-c)}{2(1+\hat{\lambda})(\alpha^2-1)} < 0.$$

(2)　Comparing the demand of each channel:

$$D_r^{GM*} - D_r^{NM*} = \frac{\alpha(1+\hat{\lambda})[\alpha\mu Q + 2(1-\mu)Q + 2k_og] + 2(1+\hat{\lambda}\alpha^2)kg + 2(1+\hat{\lambda}\alpha)s}{4[2-\alpha^2(1-\hat{\lambda})]}$$

$$\frac{+(1-\alpha)[2-\alpha^2(1-\hat{\lambda})]c - (1+\hat{\lambda}\alpha)(1-\alpha^2)c_G}{4[2-\alpha^2(1-\hat{\lambda})]} > 0,$$

$$D_o^{GM*} - D_o^{NM*} = \frac{(\mu Q + kg)\alpha + 2k_og - 2(1-\alpha^2)(c_G-s) + (1-\alpha)(2+\alpha)c}{4} > 0.$$

This completes the proof.　□

## Appendix C

**Proof of Corollary 1.** When the manufacturer does not adopt the green product supply chain, the expression of $p_r^{NM*}$ and $p_o^{NM*}$ can directly show that it is not affected by the retailer's fairness concern. Then, the first-order partial derivatives of $w^{NM*}$, $p_o^{GM*}$, $w^{GM*}$ and $p_r^{GM*}$ for $\hat{\lambda}$ are obtained, respectively.

$$\frac{\partial w^{NM*}}{\partial\hat{\lambda}} = -\frac{\mu Q - (1-\alpha)c}{2(1+\hat{\lambda})^2} < 0,$$

$$\frac{\partial p_o^{GM*}}{\partial\hat{\lambda}} = -\frac{(\mu Q + kg)\alpha^3 + 2[(1-\mu)Q + k_og]\alpha^2 - \alpha^2(1-\alpha)(2+\alpha)(c_G+s)}{2[2-\alpha^2(1-\hat{\lambda})]^2} < 0,$$

$$\frac{\partial w^{GM*}}{\partial\hat{\lambda}} = -\frac{\mu Q + kg - (1-\alpha)(c_G-s)}{2(1+\hat{\lambda})^2} < 0,$$

$$\frac{\partial p_r^{GM*}}{\partial\hat{\lambda}} = -\frac{(\mu Q + kg)\alpha^2 + 2[(1-\mu)Q + k_og]\alpha + \alpha(1-\alpha)(2+\alpha)c_G + (5\alpha^2 - 2\alpha - 2)s}{2[2-\alpha^2(1-\hat{\lambda})]^2} < 0.$$

This completes the proof.　□

**Proof of Corollary 2.** For $p_0^{NM*} - p_0^{GM*}$, $p_r^{NM*} - p_r^{GM*}$ and $w^{NM*} - w^{GM*}$, the first-order partial derivatives of $\hat{\lambda}$ are obtained:

$$\frac{\partial(p_0^{NM*}-p_0^{GM*})}{\partial\hat{\lambda}} = \frac{[(1+g)c-s]\alpha^4 + [\mu Q + kg + (1+g)c-s]\alpha^3 + [(1-\mu)Q + k_og - (1+g)c+s]2\alpha^2}{2[2-\alpha^2(1-\hat{\lambda})]^2} > 0,$$

$$\frac{\partial(p_r^{NM*}-p_r^{GM*})}{\partial\hat{\lambda}} = \frac{[(1+g)c+5s]\alpha^3 + [\mu Q + kg + (1+g)c-s]\alpha^2 + [(1-\mu)Q + k_og - (1+g)c-s]2\alpha}{2[2-\alpha^2(1-\hat{\lambda})]^2} > 0,$$

$$\frac{\partial(w^{NM*} - w^{GM*})}{\partial\hat{\lambda}} = \frac{kg + (1-\alpha)(s-cg)}{2(1+\hat{\lambda})^2} > 0.$$

It can be concluded that the first-order partial derivatives of $p_0^{NM*} - p_0^{GM*}$, $p_r^{NM*} - p_r^{GM*}$ and $w^{NM*} - w^{GM*}$ for $\hat{\lambda}$ are greater than zero. Therefore, compared with the absence of green supply chain management, the gap between product prices will increase with the increase of retailers' fairness concerns.

This completes the proof. □

**Proof of Corollary 3.** From the expression, we can directly see that the demand of each channel before adopting green supply chain management and the demand of the online direct marketing channel after adopting green supply chain management are not affected by the retailer's fairness concern:

$$\frac{\partial D_r^{GM*}}{\partial\hat{\lambda}} = \frac{\alpha(1-\alpha)(1+\alpha)[(\alpha^2-\alpha-1)(c_G-s) + (\mu Q + kg)\alpha + (1-\mu)Q + k_o g]}{2[2-\alpha^2(1-\hat{\lambda})]} > 0.$$

This completes the proof. □

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
