# Peer review of "Green Manufacturing Strategy Considering Retailers’ Fairness Concerns"

_sustainability, doi:10.3390/su11174646_

Round 1
Reviewer 1 Report
Thank you for the possibility to review this manuscript. Even as there might be contribution in this research, unfortunately my review is rather critical.
First of all, the language of the manuscript needs a thorough revision. Currently it is full of typos, poor phrase structures and incorrect and strange expressions. For example, currently it is impossible to say whether expressions such as "according to relevant data" are just a sign of poor language edit, or unwillingness to reveal correct sources.
In its current form, the literature review lacks synthesis of the literature, and is instead a list of references, reporting the approach of individual articles. This might be partly to blame for, as the key concepts are not clearly defined in the text.
Basic concepts such as Green supply chain management and Sustainable supply chain management are confused in the text. The terms should be introduced and defined with appropriate references.
Also, a question still remains, what (from the perspecitve of literature) is a dual channel? Is it a multi-channel (the description given might lean towards that direction) or omnichannel of something else? I understand the approach because of the Stackelberg game, but still as the paper is attempting to discuss supply chain management, the terms should be defined from supply chain perspective.Now th supply chain literature is rather limited.
The authors also make an attempt to refer to behavioral science, keeping the references way too superficial.
I am rather sure, that there is no such thing as “The theory of behavior science”, but the referred theory should be defined more accurately.
I also have challenges to understand the concept of "Sense of fairness", as the approach, (the distribution of income (profit) in supply chain) is also closely linked to the power-dependency framework, which is neglected.
A really major question concerning the focus of the paper, and ultimately the contribution of the paper arises:
Where is green supply chain management in the paper? Ultimately “green” is limited to a greener product. The right term might be green manufacturing, assuming that there are environmental considerations in the manufacturing process. Currently it is unclear, as the terminology varies between “Green product, environmentally friendly product, green environmental protection product, healthy green product”. In the end, there is nothing “green” in the product or the supply chain. Because of the approach, the term “green” could be replaced with anything that would increase the production cost, and/ or would have characteristics that would increase the demand.
As with the analysis, technically the authors have done a good job, even though unfortunately at the expense of theory. Because of the beforementioned challenges, and the fact that the authors introduce government subsidies in the equation, I have a hard time finding the bottom line of the paper. Can the logic (and the results) of the paper be simplified so that introducing new positive features to the product increase the profit of the supply chain in case the government subsidizes the cost increase? If so, then the result of the paper is self-evident.
Author Response
The first thing to note is that, taking into account the opinions of the review experts, this article starts from green manufacturing and modifies the article title to: Green Manufacturing Strategy Considering Retailers’ Fairness Concerns.
Point 1: First of all, the language of the manuscript needs a thorough revision. Currently it is full of typos, poor phrase structures and incorrect and strange expressions. For example, currently it is impossible to say whether expressions such as "according to relevant data" are just a sign of poor language edit, or unwillingness to reveal correct sources.
Response 1: Thank you for your comments. I have completely revised the language of the manuscript. I have made a reasonable explanation about the expression " according to relevant data" and marked the source of the data, such as: Line 52-54.
Point 2: In its current form, the literature review lacks synthesis of the literature, and is instead a list of references, reporting the approach of individual articles. This might be partly to blame for, as the key concepts are not clearly defined in the text.
Response 2: I think your comments are very important, so I thought and analyzed them carefully. According to your suggestion and combined with the characteristics of this article, change green supply chain management to green manufacturing here. The relevant literature was sorted out and compared and analyzed, and it was summarized in Section 2. And through the literature comparison table, it is more intuitive and convenient to reflect the characteristics of this article and the difference from existing research.
Point 3: Basic concepts such as Green supply chain management and Sustainable supply chain management are confused in the text. The terms should be introduced and defined with appropriate references.
Response 3: Based on your recommendations through careful analysis, in Line 29-30, here is a description of the aspects of green supply chain management, and combined with your recommendations to focus on the use of "green manufacturing" to more accurately express the important arguments of this article.
Point 4: Also, a question still remains, what (from the perspecitve of literature) is a dual channel? Is it a multi-channel (the description given might lean towards that direction) or omnichannel of something else? I understand the approach because of the Stackelberg game, but still as the paper is attempting to discuss supply chain management, the terms should be defined from supply chain perspective. Now th supply chain literature is rather limited.
Response 4: Through your comments, I reorganized the literature cited in this section. Re-referenced and replaced the literature that fits the subject of the article. Focus on the dual-channel model in which manufacturers sell products through retailer and open direct online channel to sell products. And can be seen more intuitively through Figure 1.
Point 5:The authors also make an attempt to refer to behavioral science, keeping the references way too superficial. I am rather sure, that there is no such thing as “The theory of behavior science”, but the referred theory should be defined more accurately. I also have challenges to understand the concept of "Sense of fairness", as the approach, (the distribution of income (profit) in supply chain) is also closely linked to the power-dependency framework, which is neglected.
Response 5: I think your comments are very important. So I thought and analyzed them carefully, and I refer to the existing literature to re-explain fairness concerns. Such as Line 58-59 Fairness concerns refer to a firm's concern about the inequality between supply chain parties [24]. Line 235-237, we also explain the views of Bolton [35] and Loch et al. [36] and the differences in their description of fair utility functions. In Line 238-241, referring to existing research, this article considers the fairness of retailers with reference to manufacturers' profits. Further, referring to Du et al. [41], the fairness utility function of this paper is obtained. Regarding the fair concerns arising from the impact of power on the distribution of income (profit) in the supply chain, this article is based on manufacturer-led retailers, considering the impact of retailer’s fairness. This article focuses on retailers' fairness concerns about the choice of manufacturers' green manufacturing strategies, so we do not consider the impact of fairness concerns while power conversion.
Point 6: Where is green supply chain management in the paper? Ultimately “green” is limited to a greener product. The right term might be green manufacturing, assuming that there are environmental considerations in the manufacturing process. Currently it is unclear, as the terminology varies between “Green product, environmentally friendly product, green environmental protection product, healthy green product”. In the end, there is nothing “green” in the product or the supply chain. Because of the approach, the term “green” could be replaced with anything that would increase the production cost, and/ or would have characteristics that would increase the demand.
Response 6: Your comment is very important, so I made a lot of changes to the article based on your suggestion, using "green manufacturing" to better fit the theme of this article. At the same time, the unified use of green products avoids confusion.
Point 7: As with the analysis, technically the authors have done a good job, even though unfortunately at the expense of theory. Because of the beforementioned challenges, and the fact that the authors introduce government subsidies in the equation, I have a hard time finding the bottom line of the paper. Can the logic (and the results) of the paper be simplified so that introducing new positive features to the product increase the profit of the supply chain in case the government subsidizes the cost increase? If so, then the result of the paper is self-evident.
Response 7: Based on your recommendations, reorganized in terms of theory and literature references. In order to better support this article, we have further cited actual cases. Such as: Line 64-66, the “Wuchang rice phenomenon” in china; Line 92-93, Alibaba released the "Green Action Plan" and "Green Logistics 2020 Plan"; Line 131-132, Haier sells a series of green products, such as energy conservation refrigerators and air conditioners, through their own direct channel; Line 217-219, if consumers buy purely electric vehicles such as SAIC Roewe ERX5, according to the latest subsidy policy on March 26, 2019, the total amount of actual subsidies available to consumers is 14,400 yuan; Line 395-398, battery electric vehicles (BEVs) are cheaper than the corresponding fuel vehicles (FVs): in 2016, the manufacturer's suggested retail price of a Volkswagens e-Golf was $32,157, while the basic diesel Golf was $33,226. At the same time, I also based on your suggestions, the article's overall conclusions, numerical examples of various aspects have been more accurately explained and simplified.
Reviewer 2 Report
It is a very interesting and well written paper dealing with the operation of green supply chain management by building a green dual-channel model.
My suggestions to the authors are summarized in the following comments:
· L. 82 The authors mention three streams of research GSCM. Please name the three streams
· The authors should provide information about how the proposed models can be used in real life.
Author Response
Sincerely thank you for your comments on this article.
The first thing to note is that, taking into account the opinions of the review experts, this article starts from green manufacturing and modifies the article title to: Green Manufacturing Strategy Considering Retailers’ Fairness Concerns.
Point 1: L. 82 The authors mention three streams of research GSCM. Please name the three streams. The authors should provide information about how the proposed models can be used in real life.
Response 1: Thank you for your comments. So Therefore, I have added a practical case to each stream to demonstrate the characteristics of this paper in a more inclusive way. Such as: Line 91-93, I quoted that some companies have actively extended their social responsibilities and actively tried to build green supply chains, such as: Alibaba released the "Green Action Plan" and "Green Logistics 2020 Plan". Line 130-132, many manufacturers have sold their products by opening up direct online channels. As example of the household appliance industry, Haier sells a series of green products, such as energy conservation refrigerators and air conditioners, through their own direct channel. Line 157-160, The “Wuchang rice phenomenon” is a problem caused by unfairness among members of the supply chain (the maximum selling price per kilogram of rice in Wuchang rice is 199 yuan, while rice farmers receive less than 2 yuan, and the processing cost is only 0.2 yuan).
Reviewer 3 Report
Line 57-60 authors are probably too one-sided in their approach to the subject matter, pointing to the unfairness of global (American) companies in relation to Chinese companies. Lack of neutrality and obiketivity in research. If the authors want to present examples (they should be neutral or well justified)
The general issue of honesty is treated unilaterally.
Lack of a good justification for the choice of a subject, goals described in a superficial way, lack of hypotheses,
The section on literature is superficially described as if the Authors had to write something, there is no critical approach, there are no own considerations, just a presentation of matching literature to a given topic.
Line 195-196 no justification for the market demand as a linear structure of product price and green sensitivity coefficient.
The sources of literature should be supplemented - not all are presented by the authors from the "top journals" catalogue.
Mathematical models of little use to the economy
Lack of justification for measures taken in these models
Lack of good interpretation of models
Author Response
Sincerely thank you for your comments on this article.
The first thing to note is that, taking into account the opinions of the review experts, this article starts from green manufacturing and modifies the article title to: Green Manufacturing Strategy Considering Retailers’ Fairness Concerns.
Point 1: Line 57-60 authors are probably too one-sided in their approach to the subject matter, pointing to the unfairness of global (American) companies in relation to Chinese companies. Lack of neutrality and obiketivity in research. If the authors want to present examples (they should be neutral or well justified)
Response 1: Thank you for your review. In this regard, I have re-selected new cases in China, such as: Line 64-66, The "Wuchang rice phenomenon" is a problem caused by unfairness among members of the supply chain (the maximum selling price per kilogram of rice in Wuchang Rice is 199 yuan, while rice farmers receive less than 2 yuan, and the processing cost is only 0.2 yuan).
Point 2: The general issue of honesty is treated unilaterally.
Response 2: Regarding treatment is unilateral, this article is considered this way, because the initial assumptions set in this article are based on the Stackelberg game of manufacturer leadership. In view of the fact that manufacturers occupy a leading position in the system, there is first to ensure that their profits have the first decision-making power, so they have done a unilateral treatment. At the same time, what you are proposing is also a limitation of this article. I will consider multilateral factors into it in future research.
Point 3: Lack of a good justification for the choice of a subject, goals described in a superficial way, lack of hypotheses. The section on literature is superficially described as if the Authors had to write something, there is no critical approach, there are no own considerations, just a presentation of matching literature to a given topic.
Response 3: I think your comments are very important, so I thought and analyzed them carefully. At present, in the face of environmental issues and online channel development to broaden market share and gain profits, it is necessary to consider the preferences of decision-making members, such as the example of the above-mentioned ‘Wuchang rice phenomenon’. Therefore, I have further combed the literature in this aspect, and supplemented the case's argument for this article. And compare some articles that are similar to the topic of this article, analyze the differences between this article and other articles, and make a table so that you can see the characteristics of this article more intuitively.
Point 4: Line 195-196 no justification for the market demand as a linear structure of product price and green sensitivity coefficient. The sources of literature should be supplemented - not all are presented by the authors from the "top journals" catalogue.
Response 4: Thank you for your reminder. I have been identified in this section and in this article. The literature of the reference citations has been updated and marked with new ones. References here are [20, 33-34, 40].
Point 5: Mathematical models of little use to the economy. Lack of justification for measures taken in these models. Lack of good interpretation of models.
Response 5: With regard to this part, I have re-examined carefully that the model established in this article has been demonstrated. Re-reasonable explanation and explanation of each part. And, the addition of new reasonable and effective practical cases supports this article, such as: In reality, battery electric vehicles (BEVs) are cheaper than the corresponding fuel vehicles (FVs); for example, in 2016, the manufacturer's suggested retail price of a Volkswagens e-Golf was $32,157, while the basic diesel Golf was $33,226 [42]. Due to the many changes, you can view it more carefully for your convenience. The modified ones are already in the attachment, please check.
Round 2
Reviewer 1 Report
The authors have done a lot of work in a short time to improve their manuscript. The used terminology is now more in line, and the authors have clearly made a choise to go with green manufacturing and stick ith it. What is still missing is a good an solid definition of green manufacturing. Defining the concept and introducing its dimensions thoroughly would provide the authors material to deepen their discussion and conclusions.
Of the used concepts I still have a problem with the fairness concerns. As I understand, it is still used as kind of a proxy for market power. Ultimately, it doesn't matter what the fairness concerns are, if the respective actor doesn't have any power to influence anything. So, I'm still not sure if the authors should discuss market power more than "fairness concern".
The literature review is improved, but still lacks synthesis. Yes, there are now a couple of sentences summing up the literature, but the bulk of it still consists of separate pieces of research listed one after other.
And finally, I am still sceptical that the outcome of the model is just the fat that the revenue of the system increases if somebody subsidizes the cost increase.
Author Response
Point 1: The authors have done a lot of work in a short time to improve their manuscript. The used terminology is now more in line, and the authors have clearly made a choise to go with green manufacturing and stick ith it. What is still missing is a good an solid definition of green manufacturing. Defining the concept and introducing its dimensions thoroughly would provide the authors material to deepen their discussion and conclusions.
Response 1: Thank you for your comments, which has further improved my article. About the definition of green manufacturing, in Line 91-95, I first explained the importance of green manufacturing through national manufacturing strategies such as“German industrial 4.0” and“Made in China 2025”. Then, I quoted the widely recognized definition of green manufacturing proposed by Melnyk et al. [5] in 1996. And further cited the definition of green manufacturing proposed by Pujari. [46] in 2006. It is your suggestion that also gives me some thoughts on further consolidation and improvement in the literature review section.
Point 2: Of the used concepts I still have a problem with the fairness concerns. As I understand, it is still used as kind of a proxy for market power. Ultimately, it doesn't matter what the fairness concerns are, if the respective actor doesn't have any power to influence anything. So, I'm still not sure if the authors should discuss market power more than "fairness concern".
Response 2: As you said, fairness is used as kind of a proxy for market. Actually, the reason for fair concern arises precisely because of the inequality of market forces. Therefore, I have made some deeper explanations on the reference to the literature on fair concerns. In line 169-171, proposing that due to the imbalance of market power, the distribution is uneven and the fairness is generated, and fairness concerns can influence the impact of decision-making. As proved by Choi et al. [28] experiment, individual supply chain members’ behavior shows evidence of fairness concerns for supply chain members under different power structures. Then, we analyzed the research of Cui et al. [24], retailers with fairness concerns will influence the choice of decision-making members' strategies. So, in line 184-185, we point out that this paper is similar, focusing on the impact of retailer's fairness concerns on manufacturer's green manufacturing strategy choices under dual-channle. After that, some related literatures are organized, [25] and [29-32]. In Line 198-199, we point out that this paper is handled in the same way, based on the leadership of manufacturer, and analyzes the impact of fair concerns. And from Section 4 and Section 5, it can be found that in NM mode, retailers' fair concern only affects wholesale prices; In contrast, under the GM model, the impact of retailers' fair concerns has expanded, not only affecting wholesale prices, but also affecting channel prices.
Point 3: The literature review is improved, but still lacks synthesis. Yes, there are now a couple of sentences summing up the literature, but the bulk of it still consists of separate pieces of research listed one after other.
Response 3: Thank you for your advice. It is your comments that I also found that there is a lack of comprehensiveness in the integration of the literature. Therefore, based on the previous literatures and combined with your comments above, I have further integrated this part of the literature review to better highlight the links and better support this article.
First, in terms of green supply chains (GSCs), combining the points studied in this paper: green manufacturing, the literature [5-8, 10, 17] is integrated. Then, in Line 119-120, the literature is integrated from other dimensions of the green supply chain, such as: coordination - [11], government regulation - [9, 12-13], market competition and cooperation - [15-16]. And at the end of the paragraph pointed out the difference between this article and these literature studies.
Second, regarding the dual-channel section, proposed in Line 141-144, the research is roughly divided into two streams: operation(i.e., the pricing decision, competition and cordination) under dual-channel; dual-channel structure strategy. In line 144-146, divide this article into the first stream: operation(i.e., the pricing decision, competition and cordination) under dual-channel. Then, based on these two streams, the literatures were thoroughly integrated.
Third, when synthesizing literatures of fairness concerns, I also synthesized based on your second opinion. Proving the fairness is caused by the unequal market power through Choi’s [28] experiments. Then, the literature [25, 29-32] was integrated, which is based on the manufacturer's dominant position to analyze the impact of fair-minded member decisions and the impact of the strategy. In line 198-199, we proposed that our work is treated in the same way as these research, and still use the manufacturer-led model to analyze the impact of retailer’s fairness concerns.
Point 4: And finally, I am still sceptical that the outcome of the model is just the fat that the revenue of the system increases if somebody subsidizes the cost increase.
Response 4: Regarding the subsidy part, I think the question you mentioned is very important to me. In reality, government subsidies are common, such as we proposed in Line 237-241: if consumers buy purely electric vehicles such as SAIC Roewe ERX5, according to the latest subsidy policy on March 26, 2019, the total amount of actual subsidies available to consumers is 14,400 yuan. So we can find that government subsidies are beneficial to consumers to a certain extent and can stimulate them to buy green products. This also shows that it is necessary to adopt goverment subsidies for green manufacturing development of green products. Besides, the results of the model and the numerical simulation are consistent with the actual situation. In line 529-531, we also proposed that government subsidies had a counter effect which can alleviate the negative effects of a retailer’s fairness concerns to a certain extent. This is also the same as the conclusion obtained in Zhang et al. [32]. Of course, we can also find that there are many ways for the government to promote green manufacturing and stimulate consumers to buy green products. In literature [9], the reward and punishment mechanism is adopted; literature [9] analyzed the impact of six regulation policies; in the research of Zhang et al. [32], adopting two subsidies: the fixed subsidy and discount subsidy. So this article combines the actual and reference [32], adopting government subsidy to consumers.

Reviewer 3 Report
The source is missing under Table 1.
Considerations are too mathematical. Not useful for readers who do not deal with statistical methods.
Interesting considerations, reviewer's comments taken into account
Theoretical part refined in a correct way
Author Response
Point 1:The source is missing under Table 1. Considerations are too mathematical. Not useful for readers who do not deal with statistical methods. Interesting considerations, reviewer's comments taken into account. Theoretical part refined in a correct way
Response 1: Thank you for your comments, which has helped me a lot. I have updated the source of the literature in Table 1. Regarding your comment: Consider the factors being too mathematical. I have thought deeply. This paper is similar to existing research, such as: literature [5, 7, 20, 22, 24, 32], still adopting manufacturer leadership, analyzing the impact of retailer's fairness concerns on members’ decisions and the choice of strategy under dual-channel. In line 198-199, we also emphasized this. And also combined with the actual, take the actual case to support this article. Besides, similar studies were compared, and the corresponding management implications were obtained through proof and numerical simulation. I think the reviewers have given me a lot of help and put forward effective comments for me. Help me better improve the article. In future research, I will also consider the comments of reviewers and make more successful and meaningful research.
